# Comprehensive assessment of a nationwide simulation-based course for artificial life support

**Mateusz Puslecki**[ID][1,2,3�]*, **Marek Dabrowski**[3,4☷], **Marcin Ligowski**[2☷], **Bishoy Zakhary**[5‡], **Ahmed S. Said**[6‡], **Kollengode Ramanathan**[7,8,9‡], **Elaine Cooley**[10‡], **Lukasz Puslecki**[11‡], **Sebastian Stefaniak**[2‡], **Piotr Ziemak**[12‡], **Ilona Kiel-Puslecka**[12‡], **Agata Dabrowska**[1,3], **Tomasz Klosiewicz**[1], **Maciej Sip**[1], **Radoslaw Zalewski**[1], **Malgorzata Ladzinska**[2], **Wojciech Mrowczynski**[13], **Piotr Ladzinski**[13], **Lidia Szlanga**[3], **Konrad Baumgart**[2], **Piotr Kupidlowski**[3‡], **Lukasz Szarpak**[3,14,15], **Marek Jemielity**[2], **Bartlomiej Perek**[2☷]

1 Department of Medical Rescue, Chair of Emergency Medicine, Poznan University of Medical Sciences, Poznan, Poland, 2 Department of Cardiac Surgery and Transplantology, Chair of Cardiac and Thoracic Surgery, Poznan University of Medical Sciences, Poznan, Poland, 3 Polish Society of Medical Simulation, Slupca, Poland, 4 Chair and Department of Medical Education, Poznan University of Medical Sciences, Poznan, Poland, 5 Division of Pulmonary and Critical Care Medicine, Oregon Health and Science University, Portland, OR, United States of America, 6 Division of Pediatric Critical Care Medicine, Washington University School of Medicine in St Louis and St Louis Children's Hospital, St. Louis, Missouri, United States of America, 7 Cardiothoracic Intensive Care Unit, National University Hospital, National University of Singapore, Singapore, Singapore, 8 Yong Loo Lin School of Medicine, National University of Singapore, Singapore, Singapore, 9 Bond University, Robina, QLD, Australia, 10 Extracorporeal Life Support Organization, Ann Arbor, Michigan, United States of America, 11 Department of International Management, Poznan University of Economics and Business, Poznan, Poland, 12 Center of Medical Simulation, Poznan University of Medical Sciences, Poznan, Poland, 13 Department of Pediatric Cardiac Surgery, Poznan University of Medical Sciences, Poznan, Poland, 14 Sklodowska-Curie Medical Academy, Warsaw, Poland, 15 Polish Society of Disaster Medicine, Warsaw, Poland

☷ These authors contributed equally to this work.
‡ These authors also contributed equally to this work
* mateuszpuslecki@o2.pl

**Data Availability Statement:** All assessment questionnaires before and after the course as well as tests before and after the course were made

## Abstract

### Background

Successful implementation of medical technologies applied in life-threatening conditions, including extracorporeal membrane oxygenation (ECMO) requires appropriate preparation and training of medical personnel. The pandemic has accelerated the creation of new ECMO centers and has highlighted continuous training in adapting to new pandemic standards. To reach high standards of patients' care, we created the first of its kind, National Education Centre for Artificial Life Support (NEC-ALS) in 40 million inhabitants' country in the Central and Eastern Europe (CEE). The role of the Center is to test and promote the novel or commonly used procedures as well as to develop staff skills on management of patients needing ECMO.

### Method

In 2020, nine approved and endorsed by ELSO courses of "Artificial Life Support with ECMO" were organized. Physicians participated in the three-day high-fidelity simulation-

available by ELSO as part of the accreditation as - ELSO endorsed comprehensive course. The courses assessment and results were prepared according to ELSOed education Committee. All data pertaining to the course and needed to replicate our study findings has been provided in Supplementary materials 1, 5, 6.

**Funding:** The project was awarded funding from a POWER competitive national grant (POWR.05.04.00-IP.05-00-006/18) by the Polish Ministry of Health for a total of 2,750,000 USD (PLN 10,974,708.60) (MP, MD, ML). Main reason was to develop a course about "Artificial Life Support with ECMO" offered to 264 physicians from Poland implemented in 2019-2021 at the Poznan University of Medical Sciences (PUMS). This paper includes also findings from the research project financed by the research grant of the National Science Centre (Poland) awarded based on the decision no. DEC-2015/19/D/HS4/0041 (LP). Acknowledges research support from the Children's Discovery Institute Faculty Development Award at Washington University in St. Louis (ASS). The funders had no role in study design, data collection and analysis, decision to publish, or preparation of the manuscript.

**Competing interests:** The authors have declared that no competing interests exist.

**Abbreviations:** ALS, artificial life support; CC, chest compression; CPR, cardiopulmonary resuscitation; eCPR, extended cardiopulmonary resuscitation; COVID-19, coronavirus disease; ECLS, extracorporeal life support; ECMO, extracorporeal membrane oxygenation; ELSO, Extracorporeal Life Support Organization; HEMS, Helicopter Emergency Medical Service; HPS, human patient simulator; ICU, intensive care unit; MACC, mechanical automated chest compression device; PPE, personal protective equipment; RRF, reversible respiratory failure; R&D, research and development; VV, venovenous; VA, venoarterial; DCD, donor after circulatory death; WHO, World Health Organization.

based training that was adapted to abide by the social distancing norms of the COVID-19 pandemic. Knowledge as well as crucial cognitive, behavioral and technical aspects (on a 5-point Likert scale) of management on ECMO were assessed before and after course completion. Moreover, the results of training in mechanical chest compression were also evaluated.

## Results

There were 115 participants (60% men) predominantly in the age of 30–40 years. Majority of them (63%) were anesthesiologists or intensivists with more than 5-year clinical experience, but 54% had no previous ECMO experience. There was significant improvement after the course in all cognitive, behavioral, and technical self-assessments. Among aspects of management with ECMO that all increased significantly following the course, the most pronounced was related to the technical one (from approximately 1.0 to more 4.0 points). Knowledge scores significantly increased post-course from 11.4 ± SD to 13 ± SD (out of 15 points). The quality of manual chest compression relatively poor before course improved significantly after training.

## Conclusions

Our course confirmed that simulation as an educational approach is invaluable not only in training and testing of novel or commonly used procedures, skills upgrading, but also in practicing very rare cases. The implementation of the education program during COVID-19 pandemic may be helpful in founding specialized Advanced Life Support centers and teams including mobile ones. The dedicated R&D Innovation Ecosystem established in the "ECMO for Greater Poland" program, with developed National Education Center can play a crucial role in the knowledge and know-how transfer but future research is needed.

## Introduction

The history of the use of medical simulation as a training tool dates back to the 1950s, especially in the field of emergency techniques in critical conditions. In the last 50 years of the 20th century, an intensive development of training of resuscitation techniques based on mannequin simulation were observed. Simulation for extracorporeal therapy as a tool enabling structured and comprehensive training in ECMO (extracorporeal membrane oxygenation) support was first described by Anderson in 2006 [1]. Over the past 15 years, critical state medicine simulation techniques have been refined, from low-fidelity or computer models to high-fidelity models, to mimic realistic human responses. Simulation in extracorporeal techniques has also become a powerful tool in times of crisis. In 2009, in the face of the new H1N1 pandemic, WHO recommended training based on medical simulation techniques as such that can bring the best measurable clinical outcomes [2–4].

High-fidelity simulation is a teaching methodology, a learning process that uses educational equipment from simple trainers for learning single tasks, through advanced mannequins, the so-called patient stimulators faithfully imitating the human and its parameters. The main task of medical simulation is to educate and improve patient safety. Advanced human simulators can realistically cough, vomit, and bleed. Fake blood causing medical staff real stress and the

**Table 1. Advantages of medical simulation and effective learning.**

| ADVANTAGES OF MEDICAL SIMULATION | FUNCTIONS OF HIGH-FIDELITY SIMULATION THAT LEAD TO EFFECTIVE LEARNING |
|---|---|
| • Use of real medical equipment under simulated conditions<br>• Practical exercises of invasive procedures<br>• Increasing the control of the accuracy of performed activities<br>• Constant repetition of practical skills, their assessment and analysis<br>• Enabling mistakes and showing their consequences in simulated conditions<br>• Avoiding risk to patients and learners<br>• Reduction of undesirable disturbances during exercise that may appear in the hospital or clinic<br>• The same scenario can be carried out for many students, thanks to which we achieve standardization of education<br>• Planning of clinical education based on student needs and curriculum, not patient availability<br>• Exposure to rare and complex clinical situations<br>• Drawing conclusions and summing up immediately after the end of the session during the debriefing<br>• The possibility of creating training scenarios that are very similar to real situations, thanks to which medical staff (students) can easily transfer the experience gained from training from theoretical conditions to the real situation in the future<br>• Validate the norms, standards and procedures by which student performance is assessed<br>• Possibility to diagnose educational needs on the basis of the obtained grade | • Integrating simulators into the general curriculum<br>• Clearly defined patterns and outcomes for learners using simulators<br>• Ability to repeat the skill exercises on the simulator multiple times<br>• Possibility of constructing exercises with increasing level of difficulty<br>• Ongoing feedback through the simulator during the learning process<br>• Adapting the simulator to complement a multidisciplinary teaching strategy<br>• The simulator should show the clinical variability<br>• Science should occur in a controlled environment<br>• Providing an individualized (apart from the team) educational process on the Simulator |

need for immediate action. The participants of high-fidelity medical simulation (using different techniques) can obtain better results and skills (mainly by learning-by doing) than in traditional education (Table 1) [5].

Use of ECMO as advanced life support is not a routine procedure in Poland in intensive care units, even of the highest reference. Apart from places in Poland where such treatment is carried out as part of bottom-up initiatives, there is no systemic solution. Additionally, there is lack of proper education programs that ensure easy access to this therapy [6–8]. Because the ECMO therapy organizational model is complex and expensive, we developed nationwide simulation based Artificial Life Support training in 2019 with cardiopulmonary resuscitation as an integral part of training program.

The planned educational system coincided with the emergence of the new COVID-19 pandemic. In 2020, the specter of a pandemic was confirmed, which further emphasized the legitimacy and need for systematic training in the area of extracorporeal techniques. On the other hand, the deepening epidemiological situation forced significant restrictions in practical training while maintaining the sanitary regime.

## Aim

The purpose of this study was to evaluate the results of training with the following scopes such as:

- development of medical skills and competence in the field of cardiovascular diseases, respiratory diseases resistant to conventional therapy with the additional possibility of training surgical skills necessary for ECLS,

- practical ECLS / ECMO application training, including the ability to master typical and rare emergency situations, by high fidelity simulation on clinically used equipment,

- improving the theoretical and practical skills of ECLS / ECMO in atypical, rare and difficult medical cases,

- improving trainings—supporting ECLS / ECMO competences for people with experience in this field as extended part of cardiopulmonary resuscitation (CPR),

- improving practical skills in high quality CPR, especially in chest compressions (CC),

- assessment, validation, certification and re-certification of ECLS / ECMO skills.

## Materials and methods

In 2019, we founded the "Center of Artificial Life Support and Patient Safety" within a Medical Simulation Centre of the Poznan University of Medical Sciences and then developed in the first months a project devoted to "Artificial Life Support with ECMO" as National Education Centre for Artificial Life Support (NEC-ALS). The center is equipped with modern simulation education tools and classroom technology. The training program includes theoretical and practical components related to advanced ECLS (Extracorporeal Life Support)/ ECMO therapy techniques, as well as workshops and immersive scenario-based simulations. The training program rely on valid national respiratory failure and Extracorporeal Life Support Organization (ELSO) guidelines. Of note, this is endorsed by ELSO, as first one in Poland.

The program was offered for 264 physicians specializing in anesthesia and intensive care, cardiac surgery, cardiology, thoracic surgery, vascular surgery, transplant medicine, emergency medicine, and other physicians in training from all over Poland to deploy and manage ECMO. The scope of the training was a theoretical and practical introduction to advanced extracorporeal life support (ECLS)/ECMO therapy techniques based on cardiopulmonary resuscitation guidelines, practical workshops and simulations of high fidelity. Another primary goal was to consolidate physicians' awareness regarding the availability of ECLS technology that may be used to save patients after exhaustion of conventional therapy. The program was launched in 2019 with 3 courses and ELSO endorsement. However, during the difficult period of the COVID-19 pandemic in 2020, 9 editions were carried out for 115 (course IV to XII) participants and that part of project has been analyzed. The subject matter of the course clearly corresponded to the epidemiological and training needs of the country. Most of the participants represented centers assuming the primary burden on critically ill patients as a consequence of the COVID-19 epidemic, including extracorporeal life support techniques, high-quality cardiopulmonary resuscitation, and transportation of critically ill patients.

### Participants and ethic statement

The group of 115 physicians participated in 9 editions of course in 2020 –specialists and residents in anesthesiology and intensive care, cardiology, cardiac surgery, thoracic surgery, vascular surgery, transplant medicine, and emergency medicine. The recruitment was voluntary and open to all physicians in Poland, regardless of their experience. After accepting the application, each candidate completed a study participation form and a written consent.

According to the rules of Local Bioethical Committee of Poznan University of Medical Sciences the Statement of Ethics Approval is not required.

## Form of education

The postgraduate education was formulated in—3-day course, where 50% of educational hours were implemented in the form of workshops and classes in simulated conditions—25 didactic hours spread over three days during a one weekend meeting. One week before the courses every participant was obligated to perform pre-course knowledge tests and self-assessment forms, then e-learning platform including 5 lessons of 20 slides was open for every person.

## Pre-test and pre-forms

Proposed by ELSO pre-course forms consist of demographic information's, type of physician's specialization, working place and ECMO experience and additionally 11 cognitive, 4 behavioral and 6 technical self-assessment questions (on a 5 point Likert scale), focused on aspects of managing patients supported on ECMO (S1 File). Knowledge test consists of 15 multiple choice questions on the theoretical base of artificial life support with extracorporeal techniques.

## Scope of the program

Scope of knowledge, objectives and skills are included in S2 File. The frame program (S3 File) of 3 days including first day with 6 lectures, second with water drills, priming; ultrasound visualization and 7 high-fidelity scenarios, third day including transportation workshops; 3 high-fidelity scenarios and theoretical and practice examination (S4 File). Two different ECMO platforms; Cardiohelp (Getinge, Rastatt, Germany) and Novalung (Fresenius, Bad Homburg, Germany) were used.

## Post-test and post-forms

Proposed by ELSO post-course forms consist of 11 cognitive, 4 behavioral and 6 technical self-assessment questions similar to the pre-course questions (min 1.00 and max 5.00) (S5 File). Post-course knowledge test consisted of 15 multiple choice questions on the theoretical basis of artificial life support with extracorporeal techniques (min 1.00 and max 15.00). There were no repeat questions in the pre and post-course knowledge tests. Additionally, course assessment and educational benefits were provided in 6 areas.

## Practice part

Consist of individual CC tests attempted at individual resuscitation stations Resusci Anne QCPR (Laerdal, Norway) in the course of the test A: 2-minute trial without preview–verification of the current state and B: 2-minute test with QCPR preview (quality improvement). During the test, the following parameters were monitored with Session Viewer Software 6.2.6400 (SimVentures 2019): compression rate, depth and recoil percentage. Second part consists of 15 minutes sessions of mechanical CC devices use–LUCAS (Stryker, USA). During chest compression following parameters were monitored: compression rate, depth and recoil percentage.

The last part included exam simulation scenario (pump failure in veno-arterial VA ECMO, gas supply failure on veno-venous VV ECMO), where every participant had 8 min to emergency intervention in simulated ICU room. The proposed by ELSO forms evaluated cognitive, behavioral and technical skills in addition to times to critical action completion in each scenario.

### Teaching materials

In pre-course time the 100 slides introduction presentation in e-learning form– www.ecmo.pl was opened. Additional materials included ELSO base (www.elso.org) and recommended books: "Red Book 5th Edition" (ELSO) and "ECMO Specialist Training Manual" (ELSO)– both free for students.

### ELSO endorsement and additional regulations

- As knowledge background ELSO [9] recommend for every participant Books–Red Book and ECMO Specialist Manual;

- The ELSO recommend that the course hours at least 50% should be performed in the simulated conditions;

- The necessary minimum space is 3 rooms equipped with high-fidelity simulation mannequin rooms, cannulation room and ultrasound imaging room;

- Human resources—ELSO recommends 1–2 instructors for 4 participants during practical classes. In our case, 8–10 people are involved in the simulation workshops. The entire organization of one course for 12 people requires the activity of 16 staff people: Including 4 simulation technicians, 2 ECMO specialists, 2 nurses, 4 simulation trainers and 4 doctors—instructors.

- Every program and center need ELSO accreditation–usually 1–2 control visits od ELSO Education Committee [9].

### Statistical analysis

The categorical variables were expressed as the frequency (n) and percentages (%). The quantitative variables were checked for normality distribution with the use of the Shapiro-Wilk W test. Due to the non-normal distribution among the variables, nonparametric Wilcoxon pairwise rank test was used and data were presented as median [interquartile range]. For normal distribution paired t-test for was used and data were presented as mean±SD. A value of $p < 0.05$ was considered as statistically significant. Statistical analysis was performed in R Application (1.4.1106 © 2009–2021 RStudio, PBC).

## Results

### Participants

In 2020, 115 physicians took part in the courses and all of them completed both pre and post-course documents. The predominant motivation to be trained was a will for self-improvement (n = 103), in 10 others recommendation and only in 2 superior's order. The participant demographics, specialty and hospital settings with clinical practice and ECMO experience are included in Table 2.

### Cardiopulmonary assessments

In total assessments in men activity CC depth was (56mm) in relation to women (46mm). Both groups pressed with too fast rate 120/min with not proper chest return to its original value (expansion)–chest recoil– 61.5%—Table 3.

**Table 2. Participant demographics, specialty and hospital settings with clinical practice and ECMO experience.**

| Variable | | Frequency (n) | Percent |
|---|---|---|---|
| Gender | Female | 47 | 40.9 |
| | Male | 68 | 59.1 |
| Age group | 21–30 years | 15 | 13.0 |
| | 31–40 years | 60 | 52.2 |
| | 41–50 years | 26 | 22.6 |
| | 51–60 years | 13 | 11.3 |
| | >60 years | 1 | 0.9 |
| Specialty | Other | 3 | 2.61 |
| | Physician—Anesthesiologist | 56 | 48.7 |
| | Physician—Cardiac Surgeon | 6 | 5.2 |
| | Physician—Cardiologist | 9 | 7.8 |
| | Physician—in trainee | 23 | 19.0 |
| | Physician—Intensivist | 17 | 14.8 |
| | Physician—Trauma Surgeon | 1 | 0.9 |
| | Physician—Vascular Surgeon | 1 | 0.9 |
| Hospital setting | Academic Hospital | 55 | 47.8 |
| | Community Hospital | 40 | 34.8 |
| | Government Hospital | 11 | 9.6 |
| | Other | 9 | 7.8 |
| Patient population | Adult | 94 | 81.7 |
| | Neonatal, Adult | 1 | 0.9 |
| | Pediatric | 5 | 4.3 |
| | Pediatric, Adult | 4 | 3.5 |
| | Pediatric, Neonatal | 2 | 1.7 |
| | Pediatric, Neonatal, Adult | 9 | 7.8 |
| Clinical practice duration | <5 years | 20 | 17.4 |
| | 5–10 years | 40 | 34.8 |
| | 11–15 years | 16 | 13.9 |
| | 15–20 years | 19 | 16.5 |
| | >20 years | 20 | 17.4 |
| ECMO experience | 0 months | 62 | 53.9 |
| | <6 months | 18 | 15.6 |
| | 6–12 months | 10 | 8.7 |
| | 1–2 years | 14 | 12.2 |
| | 3–5 years | 7 | 6.1 |
| | >5 years | 4 | 3.5 |
| Hospital ECMO experience | <6 months | 4 | 3.5 |
| | 6–12 months | 3 | 2.6 |
| | 1–2 years | 12 | 10.4 |
| | 3–5 years | 21 | 18.3 |
| | > 5 years | 34 | 29.6 |
| | We do not have ECMO and are not planning to | 8 | 7.0 |
| | We do not have ECMO but are planning to | 33 | 28.7 |
| Annual ECMO volume | <10 patients a year | 86 | 74.8 |
| | 10–20 patients a year | 23 | 20.0 |
| | 21–50 patients a year | 4 | 3.5 |
| | >50 patients a year | 2 | 1.7 |

*(Continued)*

**Table 2.** (Continued)

| Variable | | Frequency (n) | Percent |
|---|---|---|---|
| ECMO modes | Both VV & VA | 56 | 48.7 |
| | VA | 4 | 3.5 |
| | VV | 21 | 18.3 |
| | None | 34 | 29.6 |
| ECMO platform used | Maquet Cardiohlep | 33 | 28.7 |
| | Maquet Rotaflow | 32 | 27.8 |
| | Sorin Revolution | 4 | 3.5 |
| | Xenios Fresenius | 11 | 9.6 |
| | None | 35 | 30.4 |
| ECMO capability | Can cannulate and care for ECMO patients | 75 | 65.2 |
| | Can cannulate patients onto ECMO but then refer to outside institutions | 11 | 9.6 |
| | No ability to cannulate for ECMO but can refer to outside institutions | 20 | 17.4 |
| | No ability to cannulate or refer for ECMO | 8 | 7.0 |

## ECMO cognitive assessments

In all detailed aspects of cognitive assessment, improvement was observed after the course. Therefore, the mean value for all aspects taken together was significantly lower before (less than 4.0) than after the courses (more than 4.5) ($p<0.001$). Of note, medians of the baseline values were either 4 (in eight out of 11 aspects) or 3. After the course, all but two reached maximal value of 5 –Fig 1, Table 4.

## ECMO behavioral assessments

All courses improved significantly self-confidence of participants to play different roles in ECMO teams and also to react in emergency situations. Before training, the median value for leading ECMO team role was the lowest (1 point) whereas after the course increased by 3 points to reach 4. Mean value of all aspects of behavioral assessment increased 2-fold, from 2 to 4 ($p <0.001$)–Fig 2, Table 4.

## ECMO technical assessments

The most significant improvement in technical skills was observed after the course. Before the latter only one physician had any experience with ECMO devices. Therefore, the median values in all 6 aspects of technical experience were minimal. Following two days devoted predominantly for developing practical abilities, post-course technical assessment revealed more than 4-fold increase in expertise level in this area. Among all aspects of the aforementioned

**Table 3.** Manual chest compression assessments with and without feedback device.

| Chest compressions in CPR | Group | female | male |
|---|---|---|---|
| mean depth (mm) | 51.2±4.7 | 46.4±1.8 | 56.0±3.7 |
| mean rate (/min) | 123.3±14.3 | 124.5±17.3 | 122.2±9.7 |
| chest recoil (%) | 61.5±5.1 | 60.6±5.3 | 62.3±4.1 |
| **Chest compressions in CPR with feedback device** | **Group** | **female** | **male** |
| mean depth (mm) | 56.8±3.6 | 53.3±1.2 | 58.3±3.4 |
| mean rate (/min) | 107.8±6.7 | 104.4±6.9 | 111.3±4.5 |
| chest recoil (%) | 99.1±0.7 | 99.8±0.4 | 98.4±0.7 |

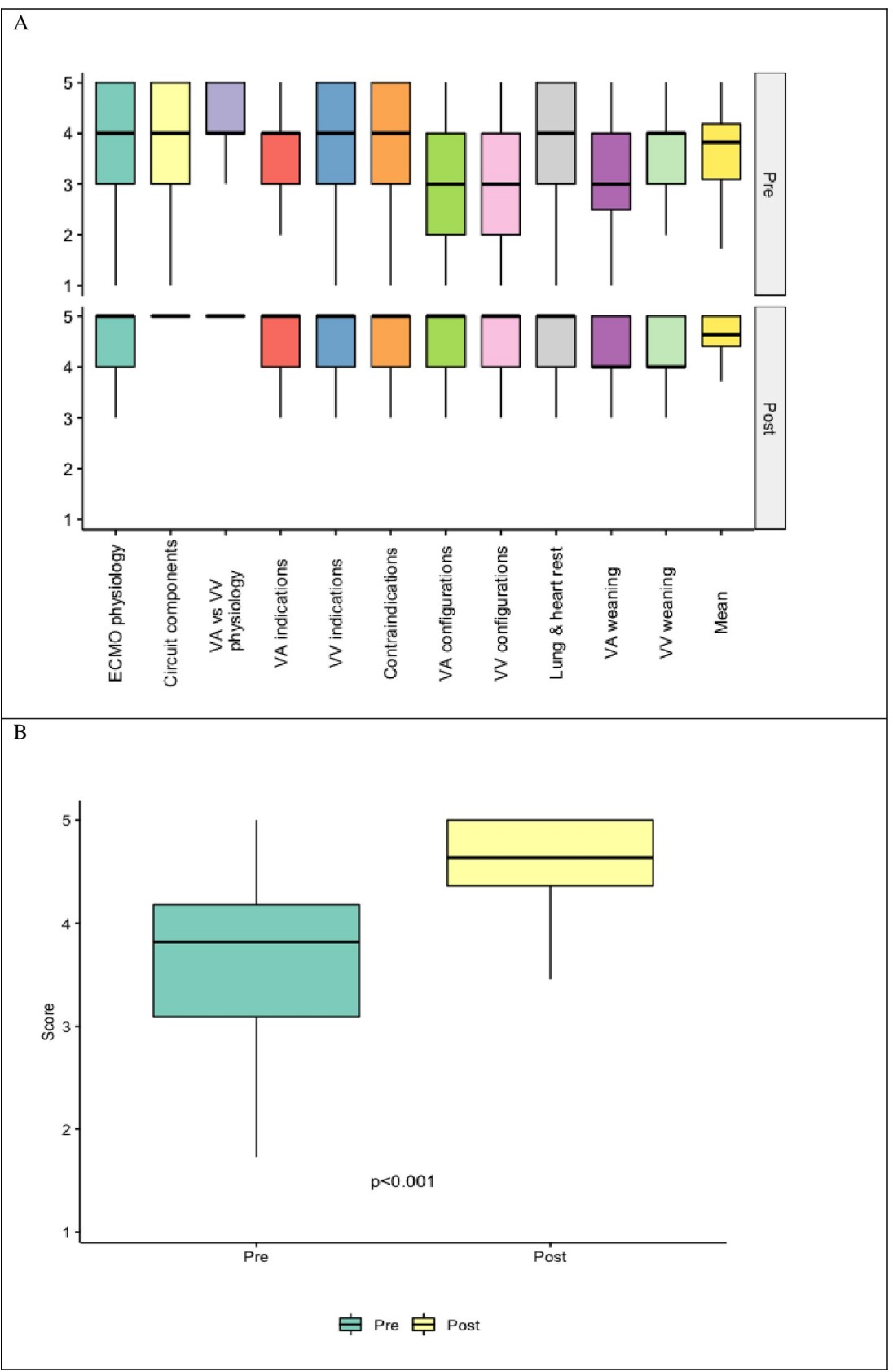

**Fig 1. Pre and post-course cognitive self-assessments.** A) Box and whisker plots of cognitive self-assessment scores by category. There was an overall increase in the median value for all assessed categories with less variability in the post-course period compared to the pre-course. B) Box and whisker plots of the mean cognitive self-assessment scores across all cognitive categories. There was a statistically significant increase in the mean post-course scores compared to the pre-course period by Wilcoxon signed-rank test (p<0.001). ECMO = extracorporeal membrane oxygenation, VV = veno-venous, VA = veno-arterial.

evaluation, the most pronounced increase was noted for management of pump failure from 1 to 5 –Fig 3, Table 4.

## ECMO knowledge assessments

It corresponded to the cognitive assessment. Pre-course knowledge regarding ECMO theory assesment tests presented mean score 11.4 out of 15 (76.0%). The results of post-course test were significantly higher than before training and mean score increased to approximately 13 points (85.3%)–Fig 4, Table 4.

## ECMO simulation assessments

In 115 group of participants, 57 completed VA pump failure sim, and rest VV gas failure scenario. The main noticed cognitive reaction was the lack of proper patient support in both scenarios–emergency ventilator settings and vasopressors.

In VA pump failure scenario the mean time to change power supply to hand crank was 200 seconds (80:450)–Fig 5A. In the VV gas failure scenario mean time to recognize gas supply problem was 55 seconds (40:115)–Fig 5B.

**Table 4. Cognitive, behavioral, technical and knowledge pre and post-course assessment.**

| Category | Variable | Pre | | | | | Post | | | | |
|---|---|---|---|---|---|---|---|---|---|---|---|
| | | *Mean* | *SD* | *Median* | *25%* | *75%* | *Mean* | *SD* | *Median* | *25%* | *75%* |
| **Cognitive** | *ECMO physiology* | 4.07 | 0.85 | 4.00 | 3.00 | 5.00 | 4.67 | 0.51 | 5.00 | 4.00 | 5.00 |
| | *Circuit components* | 3.71 | 1.11 | 4.00 | 3.00 | 5.00 | 4.77 | 0.48 | 5.00 | 5.00 | 5.00 |
| | *VA vs VV physiology* | 4.12 | 0.98 | 4.00 | 4.00 | 5.00 | 4.79 | 0.43 | 5.00 | 5.00 | 5.00 |
| | *VA indications* | 3.83 | 0.97 | 4.00 | 3.00 | 4.00 | 4.66 | 0.56 | 5.00 | 4.00 | 5.00 |
| | *VV indications* | 3.90 | 0.98 | 4.00 | 3.00 | 5.00 | 4.69 | 0.55 | 5.00 | 4.00 | 5.00 |
| | *Contraindications* | 3.84 | 1.00 | 4.00 | 3.00 | 5.00 | 4.63 | 0.60 | 5.00 | 4.00 | 5.00 |
| | *VA configurations* | 2.93 | 1.17 | 3.00 | 2.00 | 4.00 | 4.52 | 0.60 | 5.00 | 4.00 | 5.00 |
| | *VV configurations* | 3.10 | 1.16 | 3.00 | 2.00 | 4.00 | 4.66 | 0.51 | 5.00 | 4.00 | 5.00 |
| | *Lung & heart rest* | 3.67 | 1.16 | 4.00 | 3.00 | 5.00 | 4.62 | 0.57 | 5.00 | 4.00 | 5.00 |
| | *VA weaning* | 3.27 | 1.22 | 3.00 | 2.50 | 4.00 | 4.15 | 0.79 | 4.00 | 4.00 | 5.00 |
| | *VV weaning* | 3.41 | 1.19 | 4.00 | 3.00 | 4.00 | 4.30 | 0.76 | 4.00 | 4.00 | 5.00 |
| **Behavioral** | *ECMO team member* | 2.15 | 1.18 | 2.00 | 1.00 | 3.00 | 4.01 | 0.83 | 4.00 | 3.00 | 5.00 |
| | *Lead VV ECMO team* | 1.82 | 1.17 | 1.00 | 1.00 | 3.00 | 3.66 | 1.03 | 4.00 | 3.00 | 4.00 |
| | *Lead VA ECMO team* | 1.73 | 1.13 | 1.00 | 1.00 | 2.00 | 3.46 | 1.04 | 4.00 | 3.00 | 4.00 |
| | *Emergently come off ECMO* | 3.03 | 1.37 | 3.00 | 2.00 | 4.00 | 4.39 | 0.68 | 5.00 | 4.00 | 5.00 |
| **Technical** | *Circuit check* | 1.70 | 0.97 | 1.00 | 1.00 | 2.00 | 4.22 | 0.73 | 4.00 | 4.00 | 5.00 |
| | *Prime a circuit* | 1.60 | 0.97 | 1.00 | 1.00 | 2.00 | 4.18 | 0.74 | 4.00 | 4.00 | 5.00 |
| | *Safely change a circuit* | 1.40 | 0.78 | 1.00 | 1.00 | 1.50 | 3.92 | 0.86 | 4.00 | 3.00 | 5.00 |
| | *Deair a circuit* | 1.61 | 1.00 | 1.00 | 1.00 | 2.00 | 4.05 | 0.88 | 4.00 | 4.00 | 5.00 |
| | *Manage an oxygenator failure* | 1.45 | 0.83 | 1.00 | 1.00 | 2.00 | 4.16 | 0.76 | 4.00 | 4.00 | 5.00 |
| | *Manage pump failure* | 1.57 | 0.92 | 1.00 | 1.00 | 2.00 | 4.44 | 0.65 | 5.00 | 4.00 | 5.00 |
| | **Knowledge** | 11.40 | 2.81 | 12.00 | 9.25 | 13.75 | 12.82 | 1.60 | 13.00 | 12.00 | 14.00 |

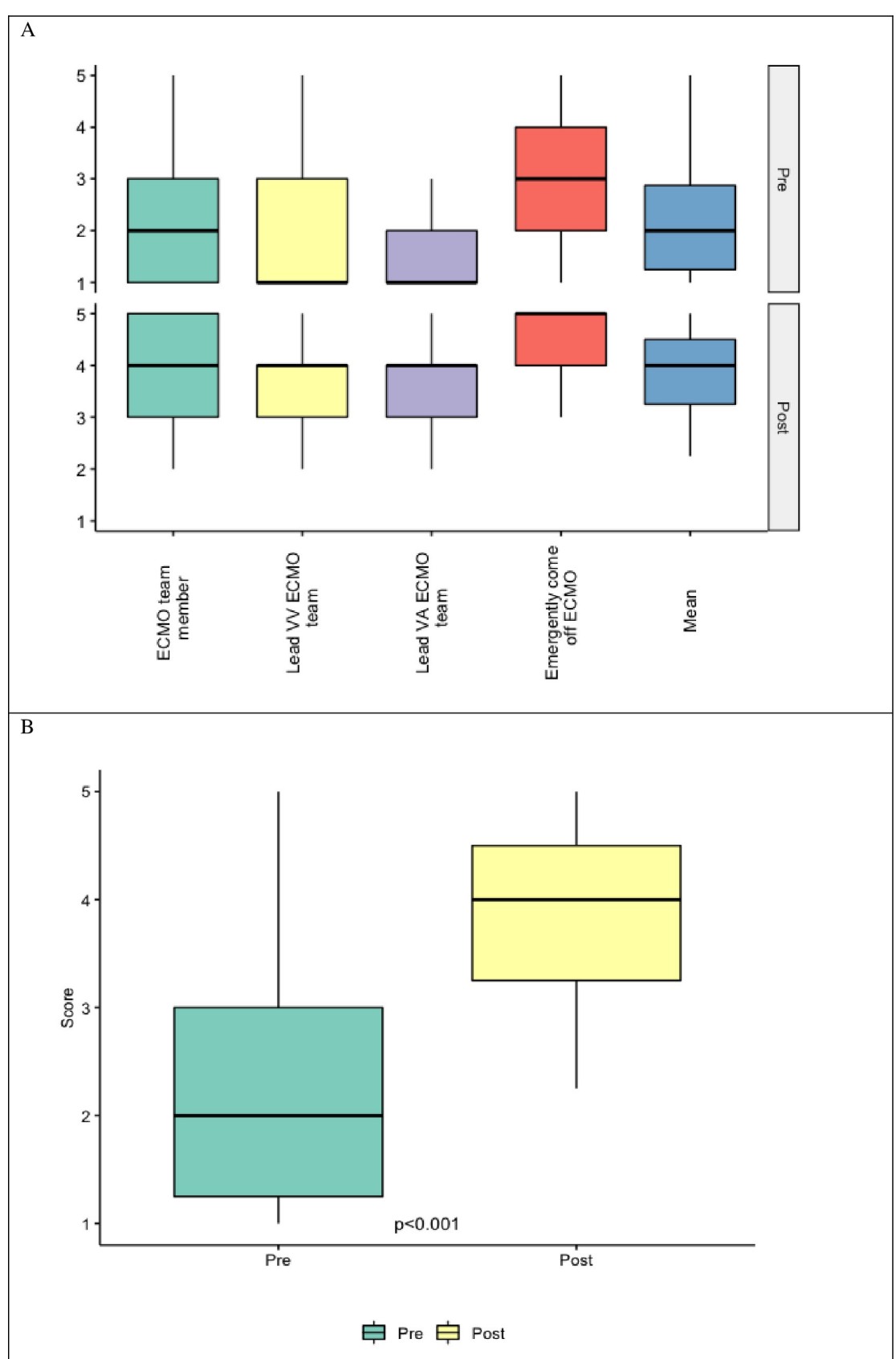

**Fig 2. Pre and post-course behavioral self-assessments.** A) Box and whisker plots of behavioral self-assessment scores by category. There was an overall increase in the median value for all assessed categories in the post-course period compared to the pre-course. B) Box and whisker plots of the mean behavioral self-assessment scores across all cognitive categories. There was a statistically significant increase in the mean post-course scores compared to the pre-course period by Wilcoxon signed-rank test (p<0.001). ECMO = extracorporeal membrane oxygenation, VV = veno-venous, VA = veno-arterial.

## Discussion

In 2020 our center was one of the few in the world, despite COVID-19 pandemic, doing courses in the online (lectures) and stationary model (hybrid). In order to adapt to COVID-19 pandemic restrictions we offered simultaneous knowledge sharing and division into smaller teams (internal innovations in COVID-19 pandemic). The relationship number of students for participants was changed–during COVID-19 pandemic: there were maximum 4 people per room and 1 instructor in theoretical part and debriefing and 2 instructors for every 2 participants in ECMO technical skills and 1 for 4 during simulations. It was possible thanks to simultaneous simulations and debriefings in three 4-person subgroups with innovative audio-visual solutions.

### Chest compressions during CPR

Chest compressions are an essential part of efficient high-quality cardiopulmonary resuscitation and ensure proper flow in the coronary and cerebral vessels. The CC quality criteria include proper rate and adequate depth of CCs, full chest recoil, and minimal interruptions during CCs and are included in European Resuscitation Council guidelines [10–13].

Despite the fact that the vast majority of participants were experienced clinicians, CC require a definite improvement. Professionals performed too fast and too superficial compressions with a low percentage of recoil. The last parameter is sometimes underestimated but it improves blood flow and maintains the compression to decompression relationship 1:1. Using CCs feedback devices improved chest compression quality in every parameter. The self-control is the most important trigger in that improvement and adaptation to the applicable guidelines.

The use of MACC allows for the systematization of CC by parameters required in the guidelines, safety patient transportation during CPR and is convenient in small rescue teams. All physicians justified to have an MACC device both in out-of-hospital conditions (emergency medical teams) and in-hospital CPR. The training influenced the achievement of the learning effect, which is the mastery of the MACC connection skills during a short 15-minute practice session, with particular attention to the correct position of the compression piston in relation to the chest [10–13].

### Extracorporeal support trainee

The issue of weaning from either VV or VA configuration was the most difficult aspect of ECMO therapy revealed by our questionnaires in cognitive and behavioral assessments. The issue of cessation (in both a positive and negative sense) of support of any failing organ is the most crucial and challenging point in the intensive care unit [14, 15]. Although, there are recommendations regarding the aforementioned situations published by different working groups and societies, there is still a lot of place for independent decision of physician responsible for patient treatment [16, 17]. These decisions are always not straightforward and many factors may have impact on the final clinical outcome. Of importance is experience in a given technology. During three-day course it was possible to extend knowledge but, in our opinion, only numerous and systematic real-life clinical applications enable to reach a true level of expertise and self-confidence (5 points).

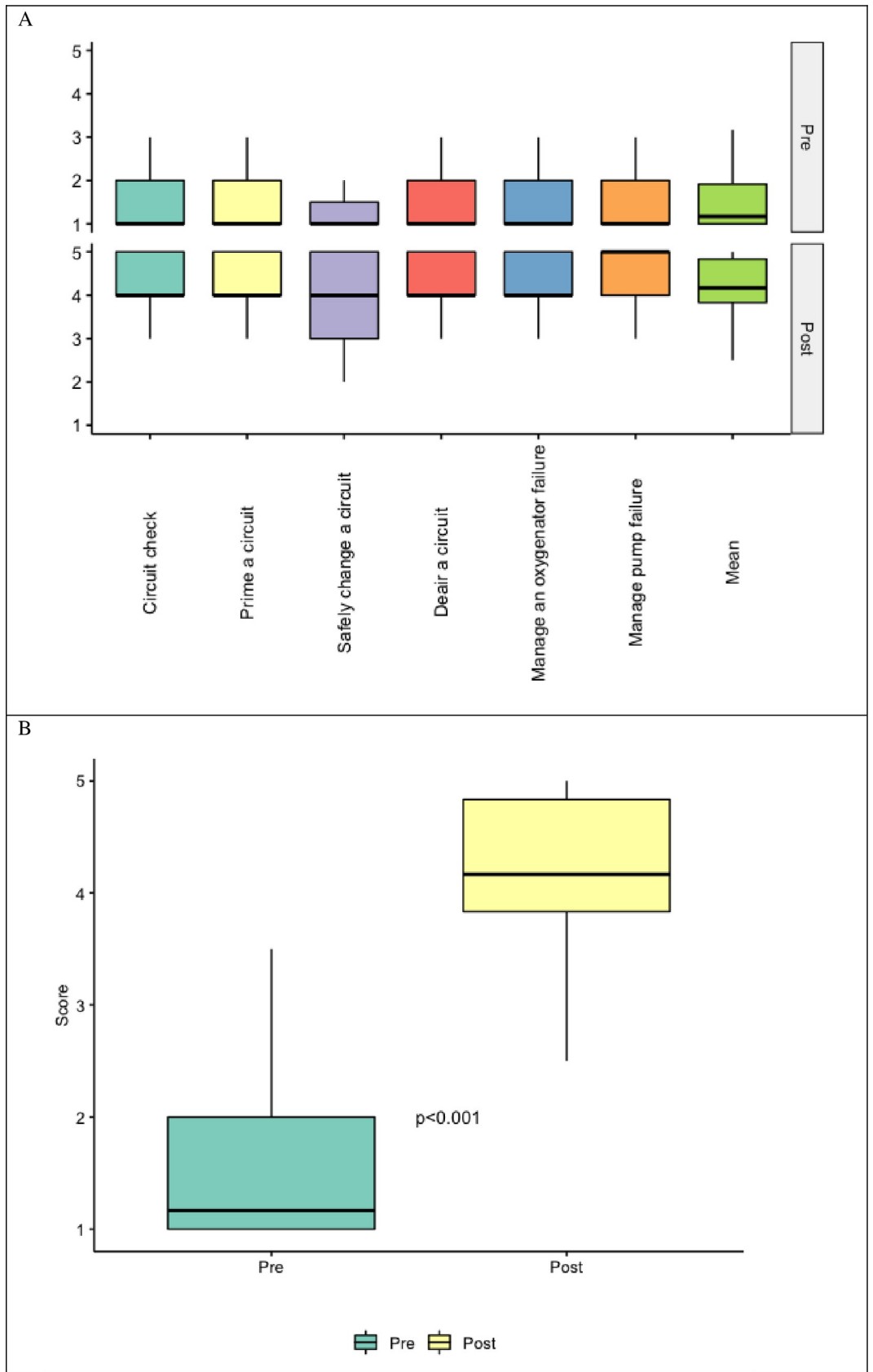

**Fig 3. Pre and post-course technical self-assessments.** A) Box and whisker plots of technical self-assessment scores by category. There was an overall increase in the median value for all assessed categories in the post-course period compared to the pre-course. B) Box and whisker plots of the mean technical self-assessment scores across all cognitive categories. There was a statistically significant increase in the mean post-course scores compared to the pre-course period by Wilcoxon signed-rank test (p<0.001).

Among behavioral aspects of management with ECMO the best improvement was revealed in playing a role of leader of the team. We considered it as great achievement of our course. One knows it is much easier to be a member than to be in charge of any novel and cutting-edge technologies. If we are immersed in ECMO therapy we are probably more willing to be a leader of team. In our opinion this improvement resulted from the confidence gained during three-day training in addition to knowledge as well as practical manual skills.

The most significant improvement after the course was observed in technical skills. There were very few physicians who had any experience with ECMO devices prior to the course. Of note, during the course development of technical skills was given utmost importance, a fact that was also reflected during technical assessment before and after training. It was also heartening to note that high fidelity simulators had an enormous educational value in ECMO ediucation in addition to being safe. [7, 8]. Management of pump failure achieved the best score after the course. It is one of the most serious catastrophes during ECMO application. Therefore, we did not only repeat many times during workshops on trouble shooting pump failure

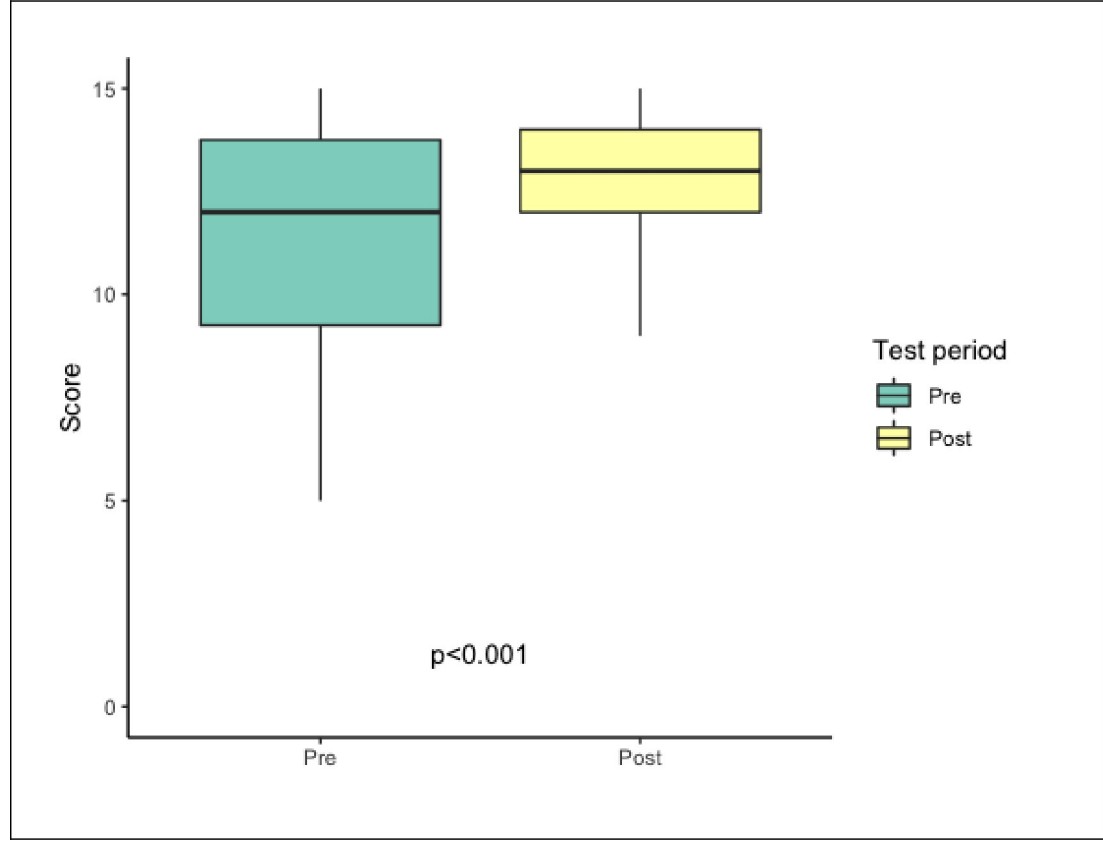

**Fig 4. Pre and post-course knowledge assessment results.** Box and whisker plots of the knowledge assessment results of the pre and post-course periods. There was a statistically significant increase in knowledge assessment scores in the post-course period (p<0.001) by Wilcoxon signed-rank test.

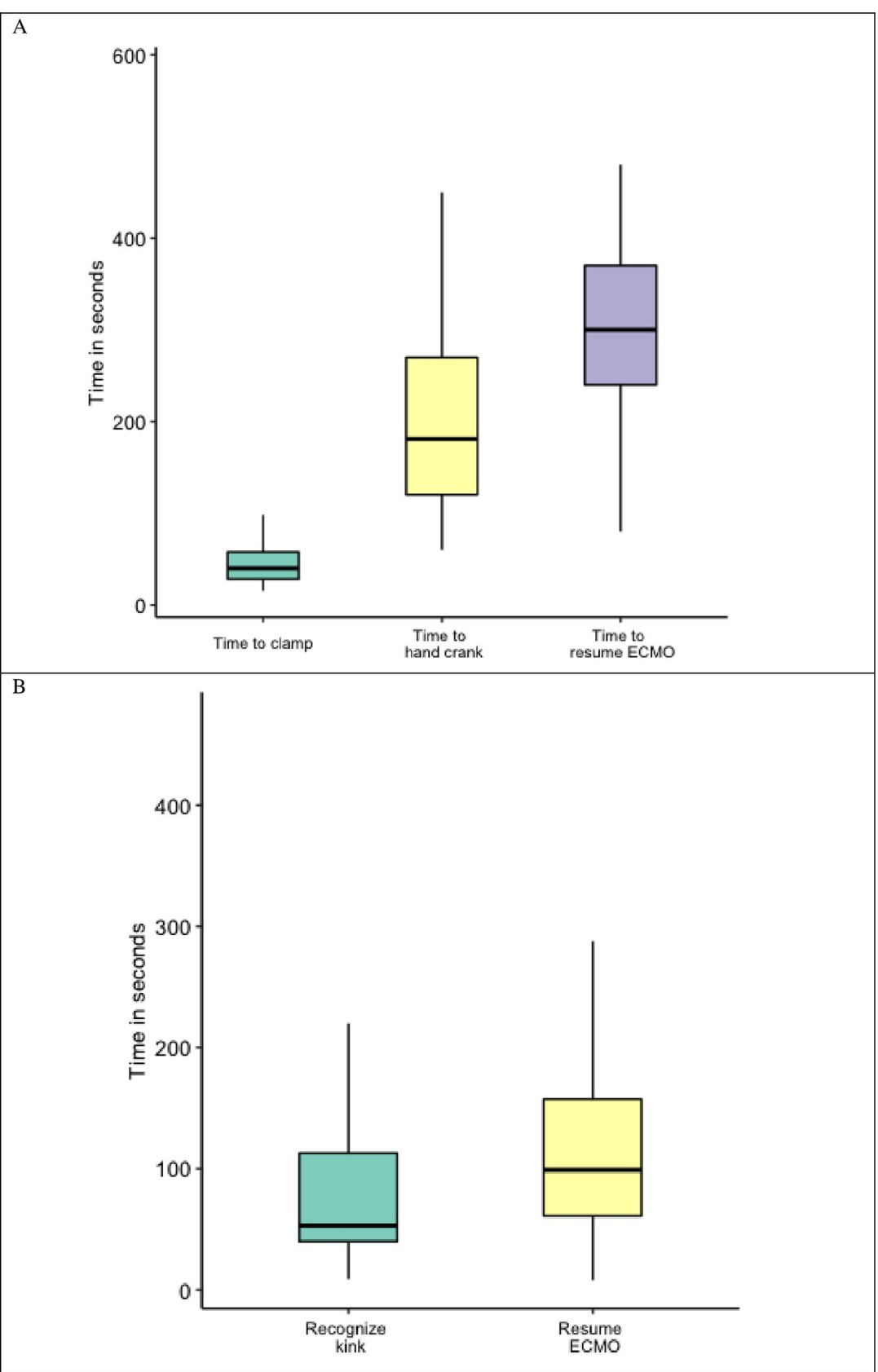

**Fig 5. Time to critical action completion in simulation scenario.** A) Box and whisker plot of time to completion of critical actions in the VA pump failure simulation scenario. B) Box and whisker plot of time to completion of critical actions in the VV gas supply failure simulation scenario. VA = veno-arterial, VV = veno-venous.

but also chose the scenario for one of the practical tests at the end of our course. The second one was the gas supply failure scenario, which was also based on real-life experience.

Knowledge assessments proved good theoretical background of participants. Pre-course knowledge regarding ECMO was more than satisfying since vast majority of physicians pointed self-improvement as the predominant motivation to participate in training. Therefore, probably most of them had read research articles or other on-line publications regarding this device before already before our course. Moreover, we cannot exclude that current pandemic and real problems with critically ill requiring invasive support made many intensivists know more about artificial life support methods. In spite of good pre-course knowledge, our training proved to be useful in imparting practical aspects for the vast majority of participating physicians.

## Translational simulation in advanced life support

In many fields, especially in cardiopulmonary resuscitation simulation techniques offers promising outcomes and have been proved in many publications. High fidelity ECMO training unfortunately have rarely been studied and published reports did not answer to the question if the use of simulators improved outcomes and/or decreased the number of serious adverse events.

Zakhary et al. presented the benefits of a practical simulation-based ECMO training in improving ECMO skills and significant reduction in delay to undertake emergency actions. This study was the first randomized controlled trial evaluating the application of simulation in ECMO education [18]. Puslecki et al. in their papers described the employment of medical simulations as a form of creating and testing tool of difficult and rare procedures. The effectiveness of translational medicine is confirmed by authors in many areas of application of extracorporeal techniques and in transporting critically ill patients. The simulation activities presented by the authors allowed for the creation of algorithms for proceedings in the regional program "ECMO for Greater Poland" [5–13, 19, 20]. Additional studies are needed to determine if simulation training will reduce complication rate and improve patients' outcomes [21–23]. The ELSO, organization established in 1989 supports health care professionals and scientists who are involved in extracorporeal techniques including guidelines creation and providing dedicated endorsed courses in artificial support subjects [24].

Patient management in critical state is definitely a team work and ECMO requires a multidisciplinary healthcare team. Sin et al. proved that simulation training that focuses on crisis resource management and proper communication is well suited to these training needs. Moreover, institutional commitment to providing public support as well as human resources is essential to the success of ECMO training programs [25]. The nurses training is crucial part in intensive care especially in sudden cardiac arrest, dynamic patient state deterioration or management with extracorporeal support. In isolated epidemic area, "hot zones" nurses are the first recipient of urgent situations and simulation training allows them to be properly prepared for such interventions [26].

In USA 46% of certified ELSO centers had an ECMO simulation program, whereas 26% report a program is in development. Sixty-three percent use simulation for summative assessment, and 76% have multidisciplinary training. Access to a simulation center allows for structured training and largest centers provide mannequin-based ECMO simulation (the most

common scenarios include pump failure (93%), oxygenator failure (90%), and circuit rupture (76%)) [27].

The most important aspect for patient with ECMO support transportation safety is a well trained and cooperating dedicated ECMO team. In Broman et al's survey, 14 of 15 centers performed programs for structured training– 33% provided full ECMO clinical and ECMO transport training, (i.e., simulations, rescue training aircraft); 60% regular ECMO simulations, 20% annual transport simulations, 33% others performed ECMO pump training and in-house simulations [20].

## Center for artificial life support and patient safety

"ECMO for Greater Poland" is an example of innovative cooperation HUB concept developed in 3.5 million inhabitant's region. Created in 2015 as nation's first, implemented all program arms to improve support in patients with severe reversible respiratory failure (RRF), hypothermia, critical states resulting in heart failure due to cardiac arrest, cardiogenic shock, or acute intoxication and promotion of donor after circulatory death (DCD) strategy in selected organ donor cases, after unsuccessful lifesaving treatment, to achieve organ recovery. In the last 4 years, many transportations of ECMO-supported critically ill patients were realized. Before launching the program, the ECMO support application in this region was incidental, despite devices availability [6–8, 19, 20].

The new sudden COVID-19 pandemic and its dynamics in 2020 accelerated the implementation of extracorporeal techniques in many countries where it had not been widely used so far. A specific group of critically ill, mechanically ventilated patients requires extended support with possible survival at VV ECMO 50–60% [24–28]. The rapid growth of teams dedicated to the treatment of patients with critical respiratory failure and the growing number of extracorporeal devices provoke the necessary training in the areas of application of extracorporeal techniques including patient transportation.

"ECMO for Greater Poland" as Artificial Life Support HUB coordinator body in the dedicated R&D Innovation Ecosystem (Fig 6) also has an educational arm of the project [20, 29–35]. The additional knowledge dissemination in the „ECMO for Greater Poland" Program is realized mainly through different scientific articles (also in interdisciplinary teams) in reputed scientific journals, presentations on different national and international conferences, as well as through organizing the international conferences „ECMO for Greater Poland" in Poznań (Poland) in 2017 and 2019, which met with high recognition in the community.

This center is unique in Poland and in the Central and Eastern Europe region–under ELSO auspicious. The authority of this organization is still improved in recent difficult year and resulted in the proper guidelines in intensive and critical care of ECMO supported patients [20]. Thanks to ELSO's trust, the project gained recognition from the Organization and during the first meeting in 2019 it was positively certified and obtained the status of ELSO-endorsed course.

Initiative was continued despite developing COVID-19 pandemic with acute epidemiological regime and participant safety in 2020 in 9 sessions for 115 physicians from whole country. In project authors' opinion it is a valuable contribution to the development of highly qualified personnel and fills the gap in the field of extracorporeal techniques in Poland.

## Development of innovation cooperation and knowledge transfer in the „ECMO for greater Poland" program in 2020

In order to improve the quality of our Educational Program we constantly monitor, evaluate and raise our competences according to current challenging times. As only one ELSO certified

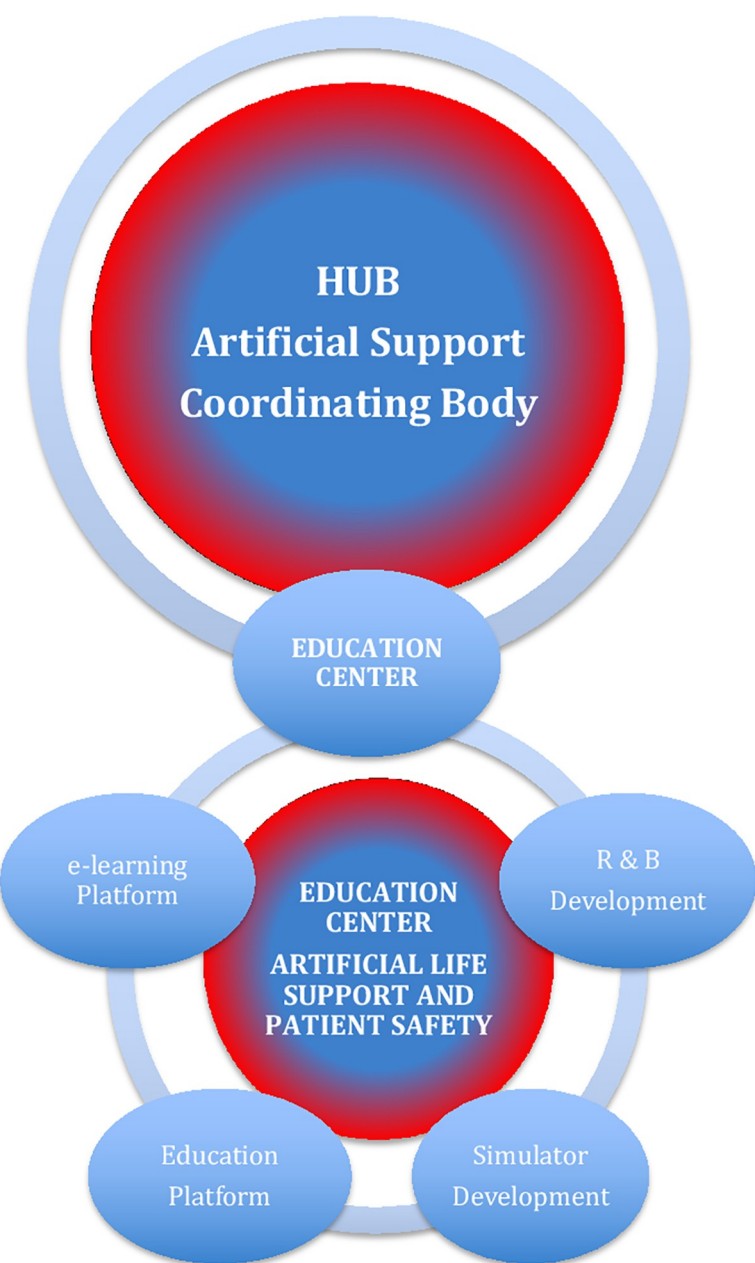

**Fig 6. R&D development: „know-how and knowledge transfer" in education center of artificial support in „ECMO for greater Poland" innovation ecosystem.**

center in Poland and in the Central and Eastern Europe we can observe huge demand for our course not only in Poland, but also in other countries (inquiries from the Central and Eastern Europe (CEE) countries like Lithuania and Latvia), especially because of COVID-19 pandemic. Despite the lockdown including of the contact trainings, the course was adapted to the epidemiological needs and in accordance with the current restrictions the 115 participants completed courses.

Nowadays we can also observe the very important result of ECMO program–popularization of extracorporeal techniques not only in Greater Poland but also in the whole country and

abroad. Such a trend is noticeable especially in developed countries, which have equipped their centers with the necessary devices for extracorporeal support over time. However, there were still obstacles in the form of a lack of separate financing, dedicated training, especially in practical skills. It was one of the reasons that's why we have decided to launch such a program and educational course. It is worth to mention the international nature of the course–because of the cooperation with worldwide organization–ELSO.

Thanks to knowledge and know-how transfer in the „ECMO for Greater Poland" program and knowledge diffusion it was possible to create new ECMO Teams in different parts of Poland, which are very desirable due to the dynamic development of the COVID-19 pandemic in our country recently. Without „ECMO for Greater Poland" Program, involvement of different stakeholders in R&D Innovation Ecosystem, as well as innovation cooperation in the whole program, it would not be possible to absorb the knowledge from the program in different part of Poland in such a short time, especially in pandemic difficult time in 2020. The devoted Education Center and Platform are a great solution in this matter–offer both teaching, learning and assessing services (according to Program Learning Outcomes) for participants of Educational Program, with ELSO support.

## Limitations

Up to now we can present only early results of our course and we must wait for at least some months or years to know how training in ALS impacted on clinical practice and ECMO applications. In spite of this limitation, the short-term results of developed concept in "Center of Artificial Life Support and Patient Safety" with ELSO cooperation are very promising. In the future, the authors together with ELSO plan to carry out a prospective assessment of the long-term outcomes in the terms of number of new and active ECMO centres in Poland, the total number of supported patients and successful transportations as well as the final outcomes of the treated individuals. Moreover, having practice in ECMO applications, we are aware how important is good cooperation with many other members of medical personnel and emergency system. This course was only for physicians. Of note, during training we appeal for participants to create their own teams and to play by them role of leaders. The significant improvement in behavioral assessment, particular leader of ECMO team, gives hope for successful involvement other medical professionals.

The publication presents early results and efficiency of simulation based training in artificial techniques. In the future, the authors together with ELSO plan to carry out a prospective assessment of the long-term outcomes in the terms of number of new and active ECMO centres in Poland, the total number of supported patients and successful transportations as well as the final outcomes of the treated individuals. Moreover, additional assessment is necessary of courses impact on the long-term ability of providers to manage patients on ECMO and the overall healthcare system or the ability to manage increasing number of patients on ECMO and the impact on patient outcomes.

The cost-consumption of the courses is their main limitation (necessary cost of expensive equipment, accommodation–hotel and meals, and recommended by ELSO books). Lack of funding in the form of a grant or a diluted government financing means that such highly sophisticated education centers are only few in the world.

## Conclusions

The dedicated R&D Innovation Ecosystem established in the "ECMO for Greater Poland" program, with developed National Education Center can play a crucial role in the knowledge and know-how transfer. More research is needed to assess the sustainability of training and its

impact on the quality of health care. The short-term results of developed concept in "Center of Artificial Life Support and Patient Safety" with ELSO cooperation are very promising.

Our course confirmed the simulation as an educational approach is valuable not only in training and testing of novel or commonly used procedures, skills upgrading, but also in practicing very rare cases. The implementation of the education program during COVID-19 pandemic may be helpful in founding specialized ECMO centers and teams including mobile ones.

## Supporting information

**S1 File. Pre course self-assessment form.**
(PDF)

**S2 File. Scope of knowledge, objectives and skills.**
(DOCX)

**S3 File. 3-day ALS course program.**
(DOCX)

**S4 File. List of immersive scenarios for ECMO therapy and corresponding learning outcomes.**
(DOCX)

**S5 File. Post course self-assessment form.**
(PDF)

**S6 File. Study data set.**
(XLSX)

## Acknowledgments

Special thanks from the authors to "ECMO for Greater Poland" Program co-authors and co-creators: Marcin Ligowski, Marek Dąbrowski, Sebastian Stefaniak, Agata Dąbrowska, Małgorzata Ładzińska, Piotr Ładziński, Konrad Baumgart, Lidia Szlanga, Tomasz Kłosiewicz, Maciej Sip, Radosław Zalewski, Piotr Ziemak, Norbert Toboła, Ilona Kiel-Puślecka, Magdalena Wieczorek, Michał Kiel, Marcin Zieliński, Aleksander Pawlak, Roland Podlewski, Lukasz Gąsiorowski, Marek Karczewski, Krzysztof Kusza, Wojciech Mrówczyński, Luksza Szarpak; Bartłomiej Perek, Marek Jemielity.

Special thanks to ELSO organization: Bishoy Zakhary, Ahmed S. Said, Kollengode Ramanathan, Elaine Cooley, Tammy Friedrich, Justyna Swol, Mark Ogino.

## Author Contributions

**Conceptualization:** Mateusz Puslecki, Marek Dabrowski.

**Data curation:** Ahmed S. Said, Kollengode Ramanathan, Elaine Cooley.

**Formal analysis:** Bishoy Zakhary, Ahmed S. Said, Kollengode Ramanathan, Sebastian Stefaniak.

**Investigation:** Ilona Kiel-Puslecka, Tomasz Klosiewicz, Radoslaw Zalewski, Konrad Baumgart, Piotr Kupidlowski, Bartlomiej Perek.

**Methodology:** Mateusz Puslecki, Marcin Ligowski, Bishoy Zakhary, Sebastian Stefaniak, Agata Dabrowska, Tomasz Klosiewicz, Malgorzata Ladzinska, Wojciech Mrowczynski, Piotr Ladzinski, Lidia Szlanga, Konrad Baumgart, Piotr Kupidlowski, Bartlomiej Perek.

**Project administration:** Mateusz Puslecki, Marek Dabrowski.

**Resources:** Lukasz Puslecki, Piotr Ziemak, Maciej Sip, Malgorzata Ladzinska, Lidia Szlanga.

**Supervision:** Marek Dabrowski, Marcin Ligowski, Bishoy Zakhary, Lukasz Puslecki, Lukasz Szarpak, Marek Jemielity.

**Validation:** Kollengode Ramanathan, Elaine Cooley, Ilona Kiel-Puslecka, Tomasz Klosiewicz, Maciej Sip, Radoslaw Zalewski.

**Visualization:** Piotr Ziemak, Ilona Kiel-Puslecka, Agata Dabrowska.

**Writing – original draft:** Mateusz Puslecki, Marek Dabrowski, Marcin Ligowski.

**Writing – review & editing:** Lukasz Puslecki, Sebastian Stefaniak, Lukasz Szarpak, Marek Jemielity, Bartlomiej Perek.

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
