## [Decision Letter · Decision Letter 0]

9 Jul 2021

PONE-D-21-14816

Efficacy of nationwide simulation based Artificial Life Support training during the COVID-19 pandemic year

PLOS ONE

Dear Authors,

Thank you for submitting your manuscript to PLOS ONE. After careful consideration, we feel that it has merit but does not fully meet PLOS ONE’s publication criteria as it currently stands. Therefore, we invite you to submit a revised version of the manuscript that addresses the points raised during the review process.

We look forward to receiving your revised manuscript.

Kind regards,

Marcel Pikhart

Academic Editor

PLOS ONE

Journal Requirements:

3. Please include additional information regarding the survey or questionnaire used in the study and ensure that you have provided sufficient details that others could replicate the analyses. For instance, if you developed a questionnaire as part of this study and it is not under a copyright more restrictive than CC-BY, please include a copy, in both the original language and English, as Supporting Information. If the original language is written in non-Latin characters, for example Amharic, Chinese, or Korean, please use a file format that ensures these characters are visible.

4. Please state whether you validated the questionnaire prior to testing on study participants. Please provide details regarding the validation group within the methods section.

6. Thank you for stating the following in the Funding Section of your manuscript:

[The project was awarded funding from a POWER competitive national grant (POWR.05.04.00-IP.05-00-006/18) by the Polish Ministry of Health for a total of 2,750,000 USD (PLN 10,974,708.60).Main reason was to develop a course about “Artificial Life Support with ECMO” offered to 264 physicians from Poland implemented in 2019-2021 at the Poznan University of Medical Sciences (PUMS). This paper includes also findings from the research project financed by the research grant of the National Science Centre (Poland) awarded based on the decision no. DEC-2015/19/D/HS4/0041.Ahmed S. Said acknowledges research support from the Children’s Discovery Institute Faculty Development Award at Washington University in St. Louis.]

 [Yes. The project was awarded funding from a POWER competitive national grant (POWR.05.04.00-IP.05-00-006/18) by the Polish Ministry of Health for a total of 2,750,000 USD (PLN 10,974,708.60). Main reason was to develop a course about “Artificial Life Support with ECMO” offered to 264 physicians from Poland implemented in 2019-2021 at the Poznan University of Medical Sciences (PUMS).]

7. We note that the grant information you provided in the ‘Funding Information’ and ‘Financial Disclosure’ sections do not match.

8. Please amend your list of authors on the manuscript to ensure that each author is linked to an affiliation. Authors’ affiliations should reflect the institution where the work was done (if authors moved subsequently, you can also list the new affiliation stating “current affiliation:….” as necessary).

9. Your ethics statement should only appear in the Methods section of your manuscript. If your ethics statement is written in any section besides the Methods, please move it to the Methods section and delete it from any other section. Please ensure that your ethics statement is included in your manuscript, as the ethics statement entered into the online submission form will not be published alongside your manuscript.

10. Please upload a new copy of Figure 6 as the detail is not clear. Please follow the link for more information: https://blogs.plos.org/plos/2019/06/looking-good-tips-for-creating-your-plos-figures-graphics/" https://blogs.plos.org/plos/2019/06/looking-good-tips-for-creating-your-plos-figures-graphics/

11. We noticed you have some minor occurrence of overlapping text with the following previous publication(s), which needs to be addressed:

- https://www.hindawi.com/journals/emi/2021/6633208/

-https://www.sciencedirect.com/science/article/abs/pii/S0735675718303127?via%3Dihub

-https://onlinelibrary.wiley.com/doi/abs/10.1111/aor.13332

In your revision ensure you cite all your sources (including your own works), and quote or rephrase any duplicated text outside the methods section. Further consideration is dependent on these concerns being addressed.

Reviewers' comments:

Reviewer's Responses to Questions

**Comments to the Author**

1. Is the manuscript technically sound, and do the data support the conclusions?

Reviewer #1: No

Reviewer #2: Yes

Reviewer #3: Yes

Reviewer #4: Yes

2. Has the statistical analysis been performed appropriately and rigorously? 

Reviewer #1: N/A

Reviewer #2: No

Reviewer #3: No

Reviewer #4: Yes

3. Have the authors made all data underlying the findings in their manuscript fully available?

Reviewer #1: Yes

Reviewer #2: Yes

Reviewer #3: Yes

Reviewer #4: Yes

4. Is the manuscript presented in an intelligible fashion and written in standard English?

Reviewer #1: Yes

Reviewer #2: Yes

Reviewer #3: Yes

Reviewer #4: No

5. Review Comments to the Author

Reviewer #1: Introduction

1

Were the aims/objectives of the study clear?

The title is somewhat complex and makes the article less interesting additionally it refers to a general concept, that dose does not reflect the aim of the paper well.

Abstract and introduction are well-formed, address the issue directly, and sufficient to the aim of the study. However, there are many typos to be revised by a native speaker.

Methods

There are many problems in the method section that threaten the validity of the study and its results:

1- There is no mention of the method used to invite the participants to this study.

2- Assessment tools used in this study are not described well: no validity test or reference to any articles that used it before.

3- ELSO is the same organization that made the assessment tool and the course guideline, which cast doubt on the validity of the assessment tools.

4- Used ELSO assessment tools are not provided.

- Sample size was fairly enough and the population was clearly defined.

- Study method was described well, which enable a researcher to repeat it.

- Main problem is that the Cardiopulmonary assessment tool is not described enough: who assesses the practical part is it the same one, who provides the feedback or not which implies a huge effect on the results.

- Which tool is used to evaluate: mean depth, mean rate (/min), chest recoil (%)

-

Result:

- The basic data was adequately described

- Analysis should be address populations that have previous experience with the study topic, which represent half of the population.

- These are immediate results of this intervention, which means we need more time to judge the real effect on clinical practice. Even though this was mentioned in the limitation section, this limitation makes this study less interesting and yields something that is already predictable.

Discussion

- Conclusions are not fully justified by the results but

- limitations of the study were discussed well.

Reviewer #2: This manuscript provided information on the utility of simulation as an educational approach in this pandemic era. Overall, this paper is well written. However following issues should be resolved to make it scientifically sound:

1. Use abbreviation with the first use of the word. Like the words Chest decompression appeared first in line 129, but abbreviated form was used from line 275. Similar use is applicable for abstract too.

2. As it was a before and after study (Quasi-experimental trial) it should be mentioned in the methodology and methodology section should be re-organized accordingly.

3. In methodology section you have mentioned Mann-Whitney U test was used in the study. But, your study design is not appropriate for this test.

4. Box and whisker plots were used in Figure 1-5. But in the results sectioned mean values were presented (But the data were not normally distributed). Use only the median (IQR) values for the presentation and analysis of data which were not normally distributed.

5. Rewrite the discussion chapter. Sections 4.3 to 4.8 are not relevant with the study, especially in the discussion part.

Reviewer #3: Interesting paper

However, please revise based on the following comments:

A. Statistical analysis:

Please revise some analysis and presentation for clarity

1. In method section,

Revised according and include all analyses conducted.

Descriptive analysis: Categorical data using frequency (n) and percentage (%). Continuous data using mean with standard deviation (SD) or median with interquartile range (IQR).

Categorical data was not included and check the spelling for "interquartile"

Pre-post test: Paired t test for normally distributed data. What is the role of Mann-Whitney U test here?

2. In result section

Table 2: please revise to frequency (n) instead if count

Rearrange again all categorical ordinal data accordingly to show the order. Example clinical practice duration, ECMO practice.

Revise the range of each category. Example for clinical practice duration , there are 5-10 years, 10-15 years and 15-20 years categories. Where do participants with practice of 10 years and 15 years belong to??

Was it?

<5

5-10

>10-15

>15-20

>20

Please check all variables i.e ECMO experience, Hospital ECMO experience, Annual ECMO volume etc

Table 3

Add the SD for all variables mean accordingly

Suggest omit Table 4 as knowledge has been presented graphically

Figure 1, 2 and 3

Omit the "mean" box-whisker in each section A since this box-whisker is repeated in Section B. Furthermore, each item of cognitive, behaviour and technical was described with median (IQR). Having the "mean" box-whisker here is confusing.

Please revise the test for Section A (median: Wilcoxon signed rank test) and Section B (mean: paired t test)

Figure 4: Revise test (mean knowledge: paired t test)

Please provide detail values [ie. mean (SD), median (IQR) etc] for cognitive, behaviour and technical (total and each item). This can be as supplementary table and include detail values of knowledge as well (Table 4). This values are important for other researchers' reference in the future.

Figure 6. not available in the manuscript

B. Discussion

The discussion is acceptable and related to the objective and findings of the study up to 4.3 only. Starting from 4.4 onwards till 4.8, it is more of a narrative review. This should be written and submitted separately.

Reviewer #4: The study meets the aims set out to evaluate the results of training.

The study design does this to some extent.

I have 3 main comments:

1. are about about what 'results' of training are expected to be - you would expect enhancement in knowledge (and cannot as the limitations note - evaluate on practice and impact until the medium or long term) - but it would be useful to know what is the comparator?

2. What was the 'input' or resource use for this training? This is important for an international readership

3. How can this be scaled up?

Minor comments:

1. I found some of the sentences hard to understand - I have highlighted some instances in the enclosed.

2. There is an excessive amount of descriptive content which can be better summarized leaving more space for analytic discussion.

Thank you for the opportunity to review this work.

6. PLOS authors have the option to publish the peer review history of their article (what does this mean?). If published, this will include your full peer review and any attached files.

Reviewer #1: **Yes: **Ahmad Altom

Reviewer #2: No

Reviewer #3: No

Reviewer #4: **Yes: **Raheelah Ahmad

---

## [Author Response · Author response to Decision Letter 0]

29 Jul 2021

Efficacy of nationwide simulation based Artificial Life Support training 

during the COVID-19 pandemic year

New title

“Comprehensive assessment of a nationwide simulation-based course 

for artificial life support”

Dear Editor and Reviewers,

Thank you very much for your valuable remarks. We really appreciate the effort of all comments to have better manuscript published. We have agreed with all of them. All recommended corrections have been described below and are highlighted (written in red) in the revised version of our manuscript.

Journal Requirements:

Response:

Manuscript was upgraded for PLOS ONE’s style requirements – Title page, manuscript, tables and figures and references.

Response:

Ethics statement: Each candidate, after accepting the application, completed a study participation form and a written consent. According to the rules of Local Bioethical Committee of Poznan University of Medical Sciences the Statement of Ethics Approval is not required.

3. Please include additional information regarding the survey or questionnaire used in the study and ensure that you have provided sufficient details that others could replicate the analyses. For instance, if you developed a questionnaire as part of this study and it is not under a copyright more restrictive than CC-BY, please include a copy, in both the original language and English, as Supporting Information. If the original language is written in non-Latin characters, for example Amharic, Chinese, or Korean, please use a file format that ensures these characters are visible.

Response:

All assessment questionnaires before and after the course as well as tests before and after the course were made available by ELSO as part of the accreditation as - ELSO endorsed comprehensive course.

Supplementary material Pre and Post assessment tools were added (Supplementary material 1 and 5). 

4. Please state whether you validated the questionnaire prior to testing on study participants. Please provide details regarding the validation group within the methods section.

Response:

The questions were developed for this purpose. They have been developed and initially tested in several US based ELSO endorsed courses before using them in the polish courses. This data is currently presented in a submitted manuscript under review for publication.

Response:

All assessment questionnaires before and after the course as well as tests before and after the course were made available by ELSO as part of the accreditation as - ELSO endorsed comprehensive course. The courses assessment and results were prepared according to ELSOed education Committee.

Data Availability – NO, all assessment questionnaires before and after the course as well as tests before and after the course were made available by ELSO as part of the accreditation as - ELSO endorsed comprehensive course. The courses assessment and results were prepared according to ELSOed education Committee.

Supplementary material Pre and Post assessment tools were added (Supplementary material 1 and 5).

6. Thank you for stating the following in the Funding Section of your manuscript:

[The project was awarded funding from a POWER competitive national grant (POWR.05.04.00-IP.05-00-006/18) by the Polish Ministry of Health for a total of 2,750,000 USD (PLN 10,974,708.60). Main reason was to develop a course about “Artificial Life Support with ECMO” offered to 264 physicians from Poland implemented in 2019-2021 at the Poznan University of Medical Sciences (PUMS). This paper includes also findings from the research project financed by the research grant of the National Science Centre (Poland) awarded based on the decision no. DEC-2015/19/D/HS4/0041.Ahmed S. Said acknowledges research support from the Children’s Discovery Institute Faculty Development Award at Washington University in St. Louis.]

 [Yes. The project was awarded funding from a POWER competitive national grant (POWR.05.04.00-IP.05-00-006/18) by the Polish Ministry of Health for a total of 2,750,000 USD (PLN 10,974,708.60). Main reason was to develop a course about “Artificial Life Support with ECMO” offered to 264 physicians from Poland implemented in 2019-2021 at the Poznan University of Medical Sciences (PUMS).]

Response:

Funding information:

The project was awarded funding from a POWER competitive national grant (POWR.05.04.00-IP.05-00-006/18) by the Polish Ministry of Health for a total of 2,750,000 USD (PLN 10,974,708.60) (MP, MD, ML). Main reason was to develop a course about “Artificial Life Support with ECMO” offered to 264 physicians from Poland implemented in 2019-2021 at the Poznan University of Medical Sciences (PUMS). 

This paper includes also findings from the research project financed by the research grant of the National Science Centre (Poland) awarded based on the decision no. DEC-2015/19/D/HS4/0041 (LP). 

Acknowledges research support from the Children’s Discovery Institute Faculty Development Award at Washington University in St. Louis (ASS).

That part is included in cover letter and was removed from manuscript.

7. We note that the grant information you provided in the ‘Funding Information’ and ‘Financial Disclosure’ sections do not match.

Response:

Funding information is included. The authors declared that no competing interests exist and there is no any financial disclosure. 

8. Please amend your list of authors on the manuscript to ensure that each author is linked to an affiliation. Authors’ affiliations should reflect the institution where the work was done (if authors moved subsequently, you can also list the new affiliation stating “current affiliation:….” as necessary).

Response:

All authors are linked to an affiliation.

9. Your ethics statement should only appear in the Methods section of your manuscript. If your ethics statement is written in any section besides the Methods, please move it to the Methods section and delete it from any other section. Please ensure that your ethics statement is included in your manuscript, as the ethics statement entered into the online submission form will not be published alongside your manuscript.

Response:

Ethic statement is only in Methods section.

Ethics statement: Each candidate, after accepting the application, completed a study participation form and a written consent. According to the rules of Local Bioethical Committee of Poznan University of Medical Sciences the Statement of Ethics Approval is not required.

10. Please upload a new copy of Figure 6 as the detail is not clear. Please follow the link for more information: https://blogs.plos.org/plos/2019/06/looking-good-tips-for-creating-your-plos-figures-graphics/" https://blogs.plos.org/plos/2019/06/looking-good-tips-for-creating-your-plos-figures-graphics/

Response:

Figure 6 file was uploaded successfully.

11. We noticed you have some minor occurrence of overlapping text with the following previous publication(s), which needs to be addressed:

-https://www.hindawi.com/journals/emi/2021/6633208/

-https://www.sciencedirect.com/science/article/abs/pii/S0735675718303127?via%3Dihub

-https://onlinelibrary.wiley.com/doi/abs/10.1111/aor.13332

In your revision ensure you cite all your sources (including your own works), and quote or rephrase any duplicated text outside the methods section. Further consideration is dependent on these concerns being addressed.

Response:

Discussion part was shorted and rephrased. All indicated publications are cited in manuscript.

Reviewers' comments:

Reviewer #1: 

Introduction

1

Were the aims/objectives of the study clear?

The title is somewhat complex and makes the article less interesting additionally it refers to a general concept, that dose does not reflect the aim of the paper well.

Response:

“Comprehensive assessment of a nationwide simulation based course for artificial life support”

Abstract and introduction are well-formed, address the issue directly, and sufficient to the aim of the study. 

However, there are many typos to be revised by a native speaker.

Response:

Manuscript was revised by a English native speaker.

Methods

There are many problems in the method section that threaten the validity of the study and its results:

1- There is no mention of the method used to invite the participants to this study.

Response:

It is included in Methods, Participants:

The recruitment was voluntary and open to all physicians in Poland, regardless of their experience. After accepting the application, each candidate completed a study participation form and a written consent.

2- Assessment tools used in this study are not described well: no validity test or reference to any articles that used it before.

Response:

All assessment questionnaires before and after the course as well as tests before and after the course were made available by ELSO as part of the accreditation as - ELSO endorsed comprehensive course. The courses assessment and results were prepared according to ELSOed education Committee. 

Supplementary material Pre and Post assessment tools were added (Supplementary material 1 and 5).

3- ELSO is the same organization that made the assessment tool and the course guideline, which cast doubt on the validity of the assessment tools.

Response:

ELSO ECMOed Taskforce was created and charged with identifying global ECMO educational needs and outlining mechanisms for international collaboration and standardization. The Taskforce listed four categories encompassing seven educational workgroups. Different workgroups created the course guidelines, the assessment tools and the certification components. The assessment tools are directly prepared by ELSO the same for every education ELSO center in the world.

Zakhary B, Shekar K, Diaz R, Badulak J, Johnston L, Roeleveld PP, Alinier G, Lai PCK, Ramanathan K, Moore E, Hassan I, Agerstrand C, Ngai WC, Salazar L, Raman L, Bembea MM, Davidson M, Gomez-Gutierrez RD, Mateo-Sidrón JAR, Kukutschka J, Antonini MV, Dickstein ML, Schmidt M, Abrams D, Ogino MT; Extracorporeal Life Support Organization (ELSO) ECMOed Taskforce. Position Paper on Global Extracorporeal Membrane Oxygenation Education and Educational Agenda for the Future: A Statement From the Extracorporeal Life Support Organization ECMOed Taskforce. Crit Care Med. 2020 Mar;48(3):406-414. doi: 10.1097/CCM.0000000000004158. PMID: 31833901.

4- Used ELSO assessment tools are not provided.

Response:

All assessment questionnaires before and after the course as well as tests before and after the course were made available by ELSO as part of the accreditation as - ELSO endorsed comprehensive course. The courses assessment and results were prepared according to ELSOed education Committee. 

Supplementary material Pre and Post assessment tools were added (Supplementary material 1 and 5).

- Sample size was fairly enough and the population was clearly defined.

- Study method was described well, which enable a researcher to repeat it.

Response: 

We thank the reviewer for the comments.

- Main problem is that the Cardiopulmonary assessment tool is not described enough: who assesses the practical part is it the same one, who provides the feedback or not which implies a huge effect on the results.

Response:

Participants performed the test A: 2-minute trial without preview – verification of the current state of Quality of CPR and B: 2-minute test with QCPR preview (quality improvement). Parameters were monitored with Session Viewer Software 6.2.6400 (SimVentures 2019).

- Which tool is used to evaluate: mean depth, mean rate (/min), chest recoil (%)

Response:

Sentence was included into Methods:

During the test, the following parameters were monitored with Session Viewer Software 6.2.6400 (SimVentures 2019): compression rate, depth and recoil percentage.

Result:

- The basic data was adequately described

- Analysis should be address populations that have previous experience with the study topic, which represent half of the population.

Response:

Use of ECMO as advanced life support is not a routine procedure in Poland in intensive care units, even of the highest reference. Apart from places in Poland where such treatment is carried out as part of bottom-up initiatives, there is no systemic solution. Additionally, there is lack of proper education programs that ensure easy access to this therapy.

- These are immediate results of this intervention, which means we need more time to judge the real effect on clinical practice. Even though this was mentioned in the limitation section, this limitation makes this study less interesting and yields something that is already predictable.

Response:

We agree that publication presents early results and efficiency of simulation based training in artificial techniques. In the future, the authors together with ELSO plan to carry out a prospective assessment of the long-term outcomes in the terms of number of new and active ECMO centres in Poland, the total number of supported patients and successful transportations as well as the final outcomes of the treated individuals.

The sentences in limitations was added:

The publication presents early results and efficiency of simulation based training in artificial techniques. In the future, the authors together with ELSO plan to carry out a prospective assessment of the long-term outcomes in the terms of number of new and active ECMO centres in Poland, the total number of supported patients and successful transportations as well as the final outcomes of the treated individuals. Moreover, additional assessment is necessary of courses impact on the long term ability of providers to manage patients on ECMO and the overall healthcare system or the ability to manage increasing number of patients on ECMO and the impact on patient outcomes.

Discussion

- Conclusions are not fully justified by the results but

- limitations of the study were discussed well.

Response:

The Conclusion part was improved:

The dedicated R&D Innovation Ecosystem established in the “ECMO for Greater Poland” program, with developed National Education Center can play a crucial role in the knowledge and know-how transfer. More research is needed to assess the sustainability of training and its impact on the quality of health care. The short-term results of developed concept in “Center of Artificial Life Support and Patient Safety” with ELSO cooperation are very promising. 

Our course confirmed the simulation as an educational approach is valuable not only in training and testing of novel or commonly used procedures, skills upgrading, but also in practicing very rare cases. The implementation of the education program during COVID-19 pandemic may be helpful in founding specialized ECMO centers and teams including mobile ones.

Reviewer #2: 

This manuscript provided information on the utility of simulation as an educational approach in this pandemic era. Overall, this paper is well written. However following issues should be resolved to make it scientifically sound:

1. Use abbreviation with the first use of the word. Like the words Chest decompression appeared first in line 129, but abbreviated form was used from line 275. Similar use is applicable for abstract too.

Response:

Thank you for that suggestion, it was corrected in manuscript.

2. As it was a before and after study (Quasi-experimental trial) it should be mentioned in the methodology and methodology section should be re-organized accordingly.

Response:

We thank the reviewer for the comments. We would like to clarify that this was not a quasi-experimental study. We wanted to assess the impact of ECMO education on the various aspects of participant’s development of technical and non-technical skills. The courses were not developed for the study. The study developed comprehensive assessments for the cognitive, behavioral and technical impacts of the course in addition to the impact of the simulation portions on the participants performance in managing ECMO emergencies. There were no study specific interventions to qualify for an experimental aspect.

3. In methodology section you have mentioned Mann-Whitney U test was used in the study. But, your study design is not appropriate for this test.

Response:

We agree, it was incorrect. That part was changed:

The categorical variables were expressed as the frequency (n) and percentages (%). The quantitative variables were checked for normality distribution with the use of the Shapiro-Wilk W test. Due to the non-normal distribution among the variables, nonparametric Wilcoxon pairwise rank test was used and data were presented as median [interquartile range]. For normal distribution paired t-test for was used and data were presented as mean±SD. A value of p<0.05 was considered as statistically significant. Statistical analysis was performed in R Application (1.4.1106 © 2009-2021 RStudio, PBC).

4. Box and whisker plots were used in Figure 1-5. But in the results sectioned mean values were presented (But the data were not normally distributed). Use only the median (IQR) values for the presentation and analysis of data which were not normally distributed.

Response:

We agree with reviewer comment. All data with not normal distribution were presented as median. Only for normal distribution date mean and SD were presented. Part of knowledge assessment was improved.

Additionaly the Table 4 was delated and we create new one for congenital, behavioral and technical assessment with knowledge assessment including mean, SD, median, min, max.

5. Rewrite the discussion chapter. Sections 4.3 to 4.8 are not relevant with the study, especially in the discussion part.

Response:

That part was changed, we removed non relevant part. We left part of importance of knowledge transfer with simulation-based training. That part can be some indication in developing Artificial Life Support centers.

Reviewer #3: 

Interesting paper

However, please revise based on the following comments:

A. Statistical analysis:

Please revise some analysis and presentation for clarity

1. In method section,

Revised according and include all analyses conducted.

Descriptive analysis: Categorical data using frequency (n) and percentage (%). Continuous data using mean with standard deviation (SD) or median with interquartile range (IQR).

Categorical data was not included and check the spelling for "interquartile"

Pre-post test: Paired t test for normally distributed data. What is the role of Mann-Whitney U test here?

Response:

We agree, that part was incorrect. That part was changed:

The categorical variables were expressed as the frequency (n) and percentages (%). The quantitative variables were checked for normality distribution with the use of the Shapiro-Wilk W test. Due to the non-normal distribution among the variables, nonparametric Wilcoxon pairwise rank test was used and data were presented as median [interquartile range]. For normal distribution paired t-test for was used and data were presented as mean±SD.. A value of p<0.05 was considered as statistically significant. Statistical analysis was performed in R Application (1.4.1106 © 2009-2021 RStudio, PBC).

2. In result section

Table 2: please revise to frequency (n) instead if count

Rearrange again all categorical ordinal data accordingly to show the order. Example clinical practice duration, ECMO practice.

Revise the range of each category. Example for clinical practice duration , there are 5-10 years, 10-15 years and 15-20 years categories. Where do participants with practice of 10 years and 15 years belong to??

Was it?

<5

5-10

>10-15

>15-20

>20

Please check all variables i.e ECMO experience, Hospital ECMO experience, Annual ECMO volume etc

Response:

Thank you for valuable suggestions. Count was changed into frequency. All categorical data were organized in order. Data frames were checked and corrected in table 2.

Table 3

Add the SD for all variables mean accordingly

Response:

According to suggestions SD for all variables mean were added and additionally in Table 4.

Suggest omit Table 4 as knowledge has been presented graphically

Response:

Table 4 was removed according to suggestions. New table with all assessments were added. 

Figure 1, 2 and 3

Omit the "mean" box-whisker in each section A since this box-whisker is repeated in Section B. Furthermore, each item of cognitive, behaviour and technical was described with median (IQR). Having the "mean" box-whisker here is confusing.

Please revise the test for Section A (median: Wilcoxon signed rank test) and Section B (mean: paired t test)

Response:

The data in section B for each figure is the mean of all the variables in each category (cognitive, behavioral and technical) compared before and after the course. We believe presenting the data in this fashion provides easier to follow presentation than a figure with comparison of the pre and post data for each variable in each category. All the data, for each variable and for the mean of all variables for each category, were not normally distributed so could only be compared using Wilcoson rank sum test and not paired t test.

Figure 4: Revise test (mean knowledge: paired t test)

Response:

For figures 1, 2 & 3 Section B compares the mean of ALL the assessed variables before and after the course. This is different data than that presented in section A. We believed comparing the before and after for each variable in each figure would be too much information and confusing for the reader so we decided to compare the Mean of each category (cognitive, technical and behavioral) before and after. For all the compared data, including the means for each category, the data were not normally distributed so was compared using Wilcoxon test instead of t test.

Similarly for figure 4, the data were not normally distributed so could not be compared by paired t test.

Please provide detail values [ie. mean (SD), median (IQR) etc] for cognitive, behaviour and technical (total and each item). This can be as supplementary table and include detail values of knowledge as well (Table 4). This values are important for other researchers' reference in the future.

Response:

The new table 4 was deleted and we create new one for cognitive, behavioral and technical assessment with knowledge assessment including mean, SD, median, min, max.

Figure 6. not available in the manuscript

Response:

Figure 6 file was uploaded successfully.

B. Discussion

The discussion is acceptable and related to the objective and findings of the study up to 4.3 only. Starting from 4.4 onwards till 4.8, it is more of a narrative review. This should be written and submitted separately.

Response:

That part of discussion was changed, we removed non relevant part. We left part of importance of knowledge transfer with simulation-based training. That part can be some indication in developing Artificial Life Support centers.

Reviewer #4:

The study meets the aims set out to evaluate the results of training.

The study design does this to some extent.

I have 3 main comments:

1. are about about what 'results' of training are expected to be - you would expect enhancement in knowledge (and cannot as the limitations note - evaluate on practice and impact until the medium or long term) - but it would be useful to know what is the comparator?

Response:

Use of ECMO as advanced life support was not a routine procedure in Poland in intensive care units, even of the highest reference before COVID pandemic. Apart from places in Poland where such treatment was carried out as part of bottom-up initiatives, there was no systemic solution. 

The overarching goal of the proposed program is: Development of knowledge and skills and consolidation of Polish doctors' awareness in the area of availability and safe use of medical technologies, which guarantee the survival of patients in the life-threatening states, in the course of acute either respiratory or circulatory failure, after exhaustion of conventional therapy as a standard of medical treatment resulting from evidence-based knowledge.

We expect that new ECMO centers will be created, new ECMO teams, or Mobile ECMO Team and total number of ECMO application will increase. We plan to evaluate these results in the survey in the future - in the period of 1 and 2 years - to assess whether new centers have been established or in centers where they were not used or are now being used. 

The long-term effect is not easy to predict. Our courses have gained importance in connection with the COVID epidemic. Total number of participants in our project is 264 physicians (it will be upgraded to 312 in 2022). The comparator of implementation of extracorporeal techniques can be: number of ECMO centers, number of ECMO applications.

Evaluation of medium- and long-term impact of the courses if very important and is the next step. We believe that the first step is to evaluate the short-term impact of the simulation courses as these are newly implemented nationwide and to our knowledge, no one has previously published the short-term impact of ECMO simulation courses. 

2. What was the 'input' or resource use for this training? This is important for an international readership

Response:

Sentences includes in Methods and Limitation

1. As knowledge background ELSO recommend for every participant Books – Red Book and ECMO Specialist Manual – it is included in Methods section.

2. The ELSO recommend that the course hours at least 50% should be performed in the simulated conditions.

3. The necessary minimum space is 3 rooms equipped with high-fidelity simulation mannequin rooms, cannulation room and ultrasound imaging room.

4. Because simulations are high fidelity we work with expensive clinical equipment, disposable equipment is also expensive.

5. Human resources - ELSO recommends 1-2 instructors for 4 participants during practical classes. In our case, 8-10 people are involved in the simulation workshops. The entire organization of one course for 12 people requires the activity of 16 staff people: Including 4 simulation technicians, 2 ecmo specialists, 2 nurses, 4 simulation trainers and 4 doctors - instructors.

6. The cost-consumption of the courses is their main limitation. Lack of funding in the form of a grant or a diluted government financing means that such highly sophisticated education centers are only few in the world.

7. Every program and center needs ELSO accreditation – usually 1-2 control visits od ELSO Education Committee. 

8. Additional cost of accommodation, hotel, meals, and recommended by ELSO books.

3. How can this be scaled up?

Response:

It was indicated in poin 2. Additionally:

1. ELSO sets a rigorous and clear time and program framework for endorsed courses and centers.

2. Our center to 2022 will perform total number of 26 courses with number 312 physicians.

3. COVID19 pandemic accelerated global interesting in artificial life support and it is not easy to estimate how it will be after pandemic time.

4. Our center is prepared to be referral education center for Central and east Europe. The development of the offer can be conducted in English and German language.

5. The development of the network of ELSO-endorsed training centers is the responsibility of ELSO. Although our initiative can be repeated anywhere, it only requires financing and development of the proposed HUB center for knowledge transfer.

Minor comments:

1. I found some of the sentences hard to understand - I have highlighted some instances in the enclosed.

Response:

According to suggestions we rephrased some sentences.

2. There is an excessive amount of descriptive content which can be better summarized leaving more space for analytic discussion.

Response:

That part of discussion was changed, we removed not relevant part. We leaved part of importance of knowledge transfer with simulation based training. That part can be some indication in developing Artificial Life Support centers.

Thank you for the opportunity to revise this work.

Mateusz Puslecki and co-authors

---

## [Decision Letter · Decision Letter 1]

25 Aug 2021

Comprehensive assessment of a nationwide 

simulation-based course for artificial life support

PONE-D-21-14816R1

Dear Authors,

We’re pleased to inform you that your manuscript has been judged scientifically suitable for publication and will be formally accepted for publication once it meets all outstanding technical requirements.

Kind regards,

Marcel Pikhart

Academic Editor

PLOS ONE

Additional Editor Comments (optional):

Reviewers' comments:

Reviewer's Responses to Questions

**Comments to the Author**

1. If the authors have adequately addressed your comments raised in a previous round of review and you feel that this manuscript is now acceptable for publication, you may indicate that here to bypass the “Comments to the Author” section, enter your conflict of interest statement in the “Confidential to Editor” section, and submit your "Accept" recommendation.

Reviewer #1: All comments have been addressed

Reviewer #2: All comments have been addressed

Reviewer #3: All comments have been addressed

2. Is the manuscript technically sound, and do the data support the conclusions?

Reviewer #1: Yes

Reviewer #2: (No Response)

Reviewer #3: Yes

3. Has the statistical analysis been performed appropriately and rigorously? 

Reviewer #1: Yes

Reviewer #2: (No Response)

Reviewer #3: Yes

4. Have the authors made all data underlying the findings in their manuscript fully available?

Reviewer #1: Yes

Reviewer #2: (No Response)

Reviewer #3: Yes

5. Is the manuscript presented in an intelligible fashion and written in standard English?

Reviewer #1: Yes

Reviewer #2: (No Response)

Reviewer #3: Yes

6. Review Comments to the Author

Reviewer #1: The Author responses were good and all the comments were addressed sufficiently.

However, a considerable changes have been added to the manuscript to make it acceptable not to mention the provided explanations.

Reviewer #2: (No Response)

Reviewer #3: (No Response)

7. PLOS authors have the option to publish the peer review history of their article (what does this mean?). If published, this will include your full peer review and any attached files.

Reviewer #1: **Yes: **Ahmad Altom

Reviewer #2: **Yes: **Dr. Farid Uddin Ahmed

Reviewer #3: No

---

## [Editor Report · Acceptance letter]

24 Sep 2021

PONE-D-21-14816R1 

Comprehensive assessment of a nationwidesimulation-based course for artificial life support 

Dear Dr. Puślecki:

I'm pleased to inform you that your manuscript has been deemed suitable for publication in PLOS ONE. Congratulations! Your manuscript is now with our production department. 

Kind regards, 

on behalf of

Dr. Marcel Pikhart 

Academic Editor

PLOS ONE